# Saint Bonaventure's Doctrine on the Virgin Mary's Immaculate Conception

José María Salvador-González 

Department of Art History, Complutense University of Madrid, 28008 Madrid, Spain; jmsalvad@ucm.es

**Abstract:** This article seeks to shed light on the approach of Saint Bonaventure of Bagnoregio (1217/21–1274) on the highly problematic issue of the Immaculate Conception of the Virgin Mary. In a context of heated debates on the matter, Saint Bonaventure presents a long and complex set of arguments that we can summarize as follows: Mary was conceived with original sin contaminating her body at first, but she was cleansed of it and sanctified immediately after her conception, at the very moment of the animation of her body, that is, when her soul gave life to her body. Therefore, the author concludes that even though the body of Mary, like that of all human beings except Christ, was conceived with original sin, it was thoroughly cleansed, and her body was sanctified from the very first moment at which it was animated by her holy soul and cleansed of all sin.

**Keywords:** Immaculate Conception; original sin; body's animation; sanctification

## 1. Introduction

Since the first centuries of our era, the Christian Church has been configuring specific essential affirmations about the person and special prerogatives of the Virgin Mary. Although at first, these affirmations were relatively restrained, the official doctrine of Christianity was expressed rather quickly in increasingly clear and exhaustive theses about the privileged status of Mary above other human beings. The heated doctrinal debates against various heresies that arose in the East during the 4th–5th centuries about the person of Jesus Christ contributed significantly to highlighting the supernatural condition of her mother, Mary. These heretical movements motivated the Church to celebrate several Ecumenical Councils in which it defined several complementary Christological and Mariological dogmas. The Council of Nicaea I (325) proclaimed as a dogma that in the unique person of Christ two different natures subsist indissolubly united: the divine and the human. Thus, when establishing this dogma, the Council of Nicaea I implicitly affirmed the thesis of the divine motherhood of Mary.

Little more than a century later, the Council of Ephesus (431) established as a dogma that the Virgin Mary is the true mother of God (Θεοτόκος), based on the thesis of the substantial union of the two different natures, the divine and the human, in the unique person of Jesus Christ. Accordingly, the Virgin Mary, being the mother of the man Christ (having engendered his human nature), is necessarily the mother of God Christ, whose divine nature is hypostatically united to his human nature in the unique person of Christ.

The Church did not take long to complement this first Mariological dogma of the divine motherhood of Mary with a second one, according to which this divine motherhood was carried out in a virginal way, without male intercourse; thus, the first dogma was completed with the declaration of the virginal, divine motherhood of Mary. In this enthusiastic dynamic in favor of Marian exaltation, the Church later established the perpetual virginity of Mary as the third Mariological dogma, meaning that she was a virgin before childbirth (*virgo ante partum*), a virgin during childbirth (*virgo in partu*), and a virgin after childbirth (*virgo post partum*).

On the contrary, two others important Mariological theses, that of the Immaculate Conception of Mary and that of her Assumption in body and soul into heaven, will be hotly debated within Christianity for many centuries. Although both theses have been accepted as pious beliefs and even as liturgical feasts since ancient times, they only became official dogmas of the Church very recently: the thesis of the Immaculate Conception became official in 1854, proclaimed by Pope Pius IX, and the thesis of the Assumption became official in 1950, proclaimed by Pope Pius XII.

This article explores just one of the personal manifestations of the medieval debates around the thesis of the Immaculate Conception of the Virgin Mary: the position assumed on this issue by the influential Franciscan teacher Saint Bonaventure of Bagnoregio (1217/21–1274), known as the Seraphic Doctor (Doctor Seraphicus).

The doctrine of Saint Bonaventure on the thesis of the Immaculate Conception of Mary has been treated in a fairly synthetic way by some authors (Pauwels 1904, pp. 44–47; Mariotti 1904, pp. 43–60; Longpré 1987, pp. 111–26; Cecchin 2001, pp. 90–91; 2003, pp. 49–53; 2005, pp. 38–54; 2008, pp. 537–38; 2021, pp. 31–33). All these authors are satisfied with summarizing in very few, brief paragraphs the extremely complex and subtle approaches of Saint Bonaventure on this hard subject. On the contrary, I believe that in this regard, the well-argued Bonaventurian position deserves a more detailed and rigorous investigation. However, to our knowledge, no study has analyzed Bonaventure's doctrine in all its complex argumentation. This is precisely the research objective I intend to fulfill in this article. Therefore, in the following extensive section, I will expose in detail the various arguments and counterarguments with which the Seraphic Doctor justifies his doctrinal position. In Section 3, I will summarize the essential conclusions that can be inferred from the reasoning of this Franciscan master.

## 2. The Position of Saint Bonaventure on the Thesis of the Immaculate Conception of Mary

The Seraphic Doctor develops his position on this issue in the three Quaestiones of Article 1 of Distinction III of the *Third Book of the Sentences of Peter Lombard*.[1] As a scholastic methodological strategy, Bonaventure first evaluates the positions of other thinkers and rejects them, or not, before proposing his personal position on the subject.

### 2.1. Evaluating the Arguments in Favor of Mary's Immaculate Conception

2.1.1. On the Eventual Sanctification of the Virgin Mary's Body before Being Animated by the Soul

In Quest. I, the author raises the problem as to whether the flesh of the Virgin Mary was sanctified before being animated. He begins by analyzing four arguments that would confirm this thesis. The first is that if Scripture affirms of Jeremiah, "Before I formed you in the womb, I knew you"; since Mary was superior to Jeremiah, it would follow that the Virgin's flesh would have been purified before being animated.[2]

The second argument is based on the testimony of Saint Elizabeth, who upon seeing her cousin Mary, proclaimed that her son exulted with joy in her womb, which would mean that the future John the Baptist, before having the spirit of life, already had the spirit of grace. Therefore, since Mary is much more worthy than John, she would have sanctification before receiving the spirit of life, that is, before being animated.[3]

The third reason is that since the sanctifying power is more powerful than the infecting power, if the flesh can be infected before being animated, it stands to greater reason that it can be sanctified before animation; if that is valid for any creature, it is even more valid for the Virgin, whose flesh would be sanctified before being animated.[4]

The fourth argument is that the Virgin Mary was conceived of holy parents and a sterile mother via the intervention of the Holy Spirit; this conception was holy and immaculate and therefore, the Virgin's flesh would have been sanctified in her very conception, before being animated.[5]

After exposing the reasons in favor of the thesis of the sanctification of the Virgin's flesh before its animation, Bonaventure explains the four arguments that would invalidate this thesis. The first, proposed by Saint Bernard, affirms that the Virgin's flesh could not be sanctified before conception since it did not yet exist, nor even at conception because of the sin that is inherent to it. The Seraphic Doctor adds to this argument that what Saint Bernard said is true not so much because of the sin of Mary's parents but because of the sin that is inherent in one's own flesh; so, it does not seem possible that Mary's flesh was sanctified before receiving the soul infusion.[6]

The second argument against the thesis is that sanctification is a gift of the grace of the Holy Spirit which is not present in the flesh but in the soul; so, it does not seem possible that grace is infused into the flesh before the soul.[7]

The third argument is that being and the perfection of grace presuppose being and the perfection of nature; therefore, if sanctification is the effect of quickening grace, it seems impossible that the flesh should be sanctified before it is quickened by the soul.[8]

As a fourth argument, Bonaventure points out that what does not concern glorification does not concern sanctification, and the flesh before being animated does not concern glorification or resurrection. For this reason, the Virgin's flesh was not sanctified before being animated.[9]

After analyzing the arguments of others for and against the thesis under study, the Seraphic Doctor exposes his personal position on the matter, taking sides for the thesis contrary to the sanctification of Mary's flesh before its animation. In his opinion, the Virgin's flesh was not sanctified before being animated, not because God could not purify Mary's flesh before animating it but because sanctification is a free gift that is added to the soul, not to the flesh. According to this, to say that the Virgin's flesh was sanctified before her animation would mean that this would be caused by the grace existing in the soul of the Virgin or in the souls of her parents. But it is impossible for this to occur through the grace existing in the soul of Mary since that would mean that her flesh was sanctified before her soul's creation when she is instead sanctified by the grace of her soul. That would mean admitting the contradiction that the before and the after exist at the same moment, which is absurd.[10]

On the contrary, to claim that the sanctification of the Virgin's flesh before being animated was caused by the grace present in the souls of her parents is also impossible. Bonaventure rejects such a possibility for three reasons. First, the grace of sanctification is not transmitted from the parents to the son since the son does not proceed from his parents according to his soul but according to his flesh (his body); therefore, Mary could not receive from her parents the grace of sanctification, which only concerns the soul.[11]

The Seraphic Doctor adduces as a second reason the fact that even if it were possible for sanctification to derive from the parents to the child, it would never derive from libidinous coitus since that would imply the contradiction that the holy and the libidinous are the same thing. Our author adds that St. Bernard makes this point when he says that holiness cannot exist without the sanctifying Holy Spirit, who, in turn, cannot be associated with sin, and sin necessarily occurs where libidinous pleasure occurs.[12]

The third reason given by Bonaventure to deny the aforementioned thesis is the following: supposing that the sanctification of Mary's flesh had been due to the absence of libido in her parents is neither convenient nor possible since only the Virgin had the exclusive privilege of conceiving without sin and giving birth without pain; this is what Saint Bernard points out when saying that the Virgin conceived of the Holy Spirit although she was not conceived of the Holy Spirit and gave birth to a virgin but was not born of a virgin. Therefore, the Seraphic Doctor concludes that the Virgin Mary's flesh was not sanctified before being animated, which he supports with four other arguments.[13]

The first is that the saying of Scripture about Jeremiah, "Before I formed you, I knew you", does not mean knowing who is formed and predestined but rather knowing the

purpose of God's predestination; from this, it does not follow that the flesh of Jeremiah or that of the Virgin was sanctified before it was formed.[14]

The second argument is that when it is said of John the Baptist that the spirit of life was not yet in him when he exulted in his mother's womb, it must be specified that in Scripture it is said that something is done when it appears and that does not exist when it does not appear. Therefore, it is said that the spirit was not in John because it did not appear since it was still in the womb, but it is said that the spirit was in him when he exulted in the womb at the coming of the Lord.[15]

As a third argument, facing the objection that the flesh can be stained and therefore, in the same way, it can also be sanctified before animation, Bonaventure argues that it is not the same in both cases since the generation of the flesh from the flesh is carnal, not spiritual; therefore, it is more normal that a vicious flesh could be derived from vicious flesh than that a holy flesh could be derived from parents sanctified by grace.[16]

Finally, the Seraphic Doctor invokes this fourth argument to deny the sanctification of the flesh of Mary. Although it is intended to say that the Virgin's conception, as the product of conjugal intercourse within a legitimate marriage with a sterile mother, lacks *actual guilt*, it does not follow that she was conceived without the *cause of sin* (that is, the original sin) because this is how the original sin of those who engender is transmitted after legitimate coitus, just as it is transmitted after adulterous coitus.[17] And it is not worth objecting that in the conception of Mary there should not be a *cause* of original sin due to the fact that she was generated by a sterile mother thanks to the power of the Holy Spirit; in that case, the same could be said of John the Baptist and of Isaac, who were also conceived by sterile mothers[18] and were both born with original sin.

Bonaventure further adds that the Holy Spirit sometimes operates as a *Spirit* and sometimes as a *Saint*. It operates as a *Spirit* when it works supernaturally, as in the conception of John the Baptist by Elizabeth and of Mary by Anne; instead, it operates as a *Saint* when it not only works supernaturally but also sanctifies what it works on, and this only happened in the conception of God the Son, the Holy of Holies, since Mary conceived of the Holy Spirit.[19]

### 2.1.2. On the Sanctification of the Virgin Mary's Soul before Contracting Original Sin

In the Second Question of Distinction 3 of Part 1, Article 1, of the *Third Book of Comments on the Sentences of Peter Lombard*,[20] Bonaventure begins by asking if the Virgin Mary's soul was sanctified before contracting original sin. To solve this question, our author follows the same methodological model as in the First Question: from the outset, he analyzes six arguments in favor and then six against before arguing his personal response to the problem.

When analyzing the first of the six arguments in favor of this thesis, the Seraphic Doctor presents the opinion of Saint Anselm, according to whom it was convenient that the conception of Jesus be made by the purest mother, of whom it was impossible to think that there was another purity higher below God. Now, the greatest purity is that in which neither actual faults nor original sin are found, and if, therefore, the Virgin was the purest, it does not seem that she contracted original sin, for which reason she seems to have been sanctified before contracting original sin.[21]

The second favorable argument is based on St. Augustine, who affirms that he does not want to raise questions about sins with respect to the Mother of the Lord, which seems to indicate that there were no sins committed or contracted in her (original sin).[22]

The third reasoning is supported by St. John Damascene when he says that, "The honor of the mother is reverted to the Son"; now, all honor is due to the Son of the Virgin, especially in terms of his immunity from sin; therefore, if it corresponds to the honor of the Son to have a pure and holy mother, it seems that the Mother of Christ would be immune from all sin, both original and current.[23]

The fourth argument in favor of the immaculate thesis is based on the assumption that there is always a middle ground between the extremes. In this sense, compared to those who have original sin in the soul and in the flesh, as in the children of Adam, there is someone, Christ, who lacks original sin in the soul and in the flesh; therefore, between both extremes there must be someone who, as a middle ground, has original sin either in the flesh, but not in the soul, or in the soul, but not in the flesh. Now, since it is impossible to have sin in the soul and not in the flesh, since the original birth comes from the flesh because only the flesh is transmitted, it remains only—as the Seraphic Doctor concludes—that someone has original sin only in the flesh and not in the soul: this is the Virgin Mary, who maximally acceded to Christ in all purity.[24]

The fifth favorable argument maintains that the sanctification of the Virgin surpasses that of all other saints; therefore, since St. John the Baptist was sanctified at birth *from the womb* (ex utero), it appears that the Virgin Mary was not only sanctified thus but also at birth *in the womb* (in utero). But, as birth in the womb occurs in the infusion of the soul, the Virgin therefore had the sanctity of divine grace at the very moment of the infusion of her soul, and so she was never infected with original sin.[25]

The sixth argument in favor of Mary's Immaculate Conception argues that it was possible to give grace to the soul of the Virgin in the first moment of its creation; but, since it was convenient for God to give this soul what was convenient for it to receive, it seems that in the first moment of its creation, he gave it original grace. Therefore, since grace cannot coexist with sin, either she lost grace or did not contract original sin. But since it cannot be said that she lost grace, it seems that it must be said that she did not have original sin.[26]

### 2.2. Evaluating the Reasons Contrary to the Thesis of Mary's Immaculate Conception

After explaining the arguments in favor of Mary's Immaculate Conception, Bonaventure analyzes the six arguments against this thesis. The first is based on the epistle to the Romans when Saint Paul affirms that "all sinned in Adam" because we all exist seminally in Adam; since the Virgin Mary also existed seminally in Adam, she also contracted original sin, like all the others.[27]

The second proof against the thesis is based on the sentence of Saint Augustine that "No one is freed from sins except by faith in the Redeemer"; therefore, all are saved by Christ. However, since someone who is without sin (that is, Christ) is not freed from it, it turns out that all others who are not Christ contracted original sin from which they would be freed by Christ.[28]

The third demonstration contrary to the above thesis is based on the position of Saint Bernard, who says that the Savior, when he came to free us all, found no one who was free from sin; therefore, he also did not find that the Virgin was free, which means that she contracted original sin.[29]

In the fourth proof against Mary's Immaculate Conception, Bonaventure builds this complex reasoning. If the Virgin lacked original sin, she would have lacked the death penalty that is inherent to her; therefore, either she was done a great injustice, since she died, or she would have died in a dispensative way for the salvation of humanity. But the first alternative would be an *outrage to God* because then God would not be a just retributor, and the second alternative would be an *outrage for Christ*, because then Christ would not be a sufficient Redeemer. Thus, since both alternatives are false and impossible, it only remains to admit that Mary contracted original sin.[30]

As a fifth argument contrary to the above thesis, the Seraphic Doctor argues that only those who are guilty benefit from Christ's redemption, and if the Virgin lacked original sin, she would not benefit from Christ's redemption. And, since it is a great glory for the saints to have been redeemed by Christ, if he had not redeemed the Virgin, she would have been deprived of the nobliest glory, which is a profane and impious statement.[31]

Finally, Bonaventure formulates the sixth argument against the immaculist thesis: if the Virgin Mary did not have original sin, and knowing that only Christ opened the door of heaven that original sin closed us, it would happen that, if Mary had died before Christ,

she would have immediately flown to heaven. This would make it absurd that Saint Paul's sentence to the Colossians was false when he said that "Christ pleased to reconcile by himself everything that exists in heaven or on earth."[32]

*2.3. Bonaventure's Position against Mary's Immaculate Conception*

After exposing the arguments for and against the mentioned thesis, the Seraphic Doctor argues his personal position on the matter, namely, affirming, in accordance with the common opinion at this time, that the Virgin's sanctification occurred after contracting original sin.[33]

2.3.1. Justifying His Opinion Contrary to the Thesis of Mary's Immaculate Conception

Now, before reasoning out his contrary position in this matter, our author once again carefully revisits the arguments of those who defend the Immaculate Conception of Mary.

In this order of ideas, Bonaventure begins by saying that some affirmed that in the Virgin's soul at her birth, the grace of sanctification came before original sin. This is justified, according to immaculists, because it was convenient that at birth, the soul of the Virgin should be sanctified with the highest excellence above all other saints, and this excellence should be not only in terms of the *abundance* of holiness but also in terms of the *acceleration* of time so that at the moment of its creation, grace was infused, and at this very moment, its soul was infused into her flesh (body).[34] The Seraphic Doctor complements the reasoning of the defenders of the Immaculate Conception, adding that since Wisdom is more mobile than all beings that move, and grace is more powerful than nature, the effect of the grace of holiness prevailed in the flesh of Mary more than the effect of the stain on the soul; from this, it follows that she did not contract original sin.[35]

Bonaventure adds that the position of the defenders of the Immaculate Conception seems to be based on multiple congruences, namely, because of *the extraordinary honor of Christ*, to whom it was fitting to be born of a most pure mother, because of *the exclusive prerogative of the Virgin*, who had to excel in sanctification above the other saints, and finally, for *the honor of the order.* Thus, just as there was a person (Christ) immune to original sin in his flesh and in his soul and in its cause and effect, and as there were people (humankind as a whole) who, on the contrary, contracted original sin in the flesh and in the soul, so too, the immaculists conclude, should there be an average person who somehow had or somehow did not have original sin: that average person is the Virgin Mary, the mediator between Christ and us, just as Christ is the mediator between God and us.[36]

Our author then highlights that the defenders of the Immaculate Conception rely on the testimony of Saint Anselm when he says that the Virgin was the purest, with a purity that cannot be conceived greater than that below God.[37] In addition, the Seraphic Doctor points out that those in favor of the immaculate thesis claim that not only does it not oppose but it agrees with *the truth of Holy Scripture* and *the truth of the Christian faith*.[38] It agrees with *the truth of Holy Scripture* because the Virgin was prefigured by the *ark of the covenant*, and her soul was prefigured by *the urn* in which the manna was deposited; since *the urn* was filled with manna before being placed in the *ark of the covenant*, this means that naturally, the Virgin's soul was sanctified before being united to her flesh.[39] They also affirm that their immaculist thesis does not contradict the *Christian faith* either when they say that the Virgin was freed from original sin by grace. However—as the defenders of the immaculate thesis point out—although this grace prevented the infection of Mary's soul, it did not prevent the *infection of her flesh*, and due to that *infection of her body*, the penalties were maintained in the Virgin, since the grace of sanctification does not avoid *penalties* but *guilt*.[40] And, although Mary was subjected to hardships, she was nevertheless freed by Christ from original sin in a very different way from the others, since the others were raised after the fall (sin), while the Virgin was sustained on the Fall so that she would not stumble, that is, Christ freed her from original sin. Thus, the defenders of the immaculate doctrine conclude that the Virgin was not infected by original sin as to *the effect* but only as to *the cause*.[41]

Now then, in the face of the allegations of the immaculate, Saint Bonaventure presents the arguments of those who, like him, believe that the sanctification of Mary occurred after contracting original sin, like all other human beings except her divine Son, Christ, the only one immune to original sin.[42] This is what Saint Paul affirms in his epistle to the Romans, when he says that "All have sinned and need the glory of God". This means that the grace of Christ, the only one who was born without original sin, comes to all sinners, and all need of the glory of God so that Christ frees them from sins.[43] Saint Augustine expresses himself in the same sense when, commenting on John the Baptist's phrase defining Jesus as "the lamb of God who takes away the sins of the world", he says that only the one who came without sin and lacks of sin can take away the sins of the world.[44]

The Seraphic Doctor later ensures that the opinion denying Mary's Immaculate Conception is *more common*, *more reasonable*, and *more secure* than its contrary. It is *more common*, because almost everyone maintains that the Virgin had original sin, as shown by the multitude of her penalties; in this regard, the author points out that there is no reason to say that Mary suffered these penalties to redeem others but rather because she contracted original sin.[45] He then states that the anti-immaculate opinion is *more reasonable* than the opposite because the being of nature temporarily and naturally precedes the being of grace, as Saint Augustine suggests when he affirms that "to be born is first before being reborn". In the same sense, it is said that first is *to be* before *being good*; therefore, it is first necessary that the soul be united with the flesh before the grace of God is infused into it. If, therefore, Mary's flesh was infected by original sin, her soul would also be infected by that infection by original sin. For this reason, our author considers it necessary to affirm that the infection of original sin in Mary occurred before her sanctification.[46]

Third, Bonaventure emphasizes the idea that the anti-immaculate opinion is also *safer* than its opposite because it agrees *with the piety of the faith* and with *the authority of the saints*. In his opinion, it agrees with the authority of the saints because they commonly maintain that only Christ is exempt from original sin since "All sinned in Adam", and no saint has been heard to affirm that the Virgin was immune to original sin.[47] The author takes his reasoning to the point of ensuring that the anti-immaculate position is even more consistent with *the piety of the faith*. This is confirmed, according to him, because although the Mother of Christ should be the object of our reverence and devotion, this reverent devotion should be given with greater reason to Christ: all honor and all glory should be given to Him alone since He is the *Redeemer* and Savior of all who *opened the door* of heaven to all of us and died for all of us, including the Virgin Mary herself.[48]

In the end, the Seraphic Doctor opts in this problem to adhere to "the common opinion", which maintains that the Virgin was sanctified after contracting original sin, considering that this contraction does not diminish her honor, which in any case, is incomparably less than that of his divine Son Jesus.[49] To support his position, Bonaventure provides these six arguments.

1.  First, he refutes the thesis that the Virgin was pure based on these three reasons. First, because this thesis says that the supreme purity of the Virgin is *below God*, it means that it is inferior to the purity of Christ; therefore, Mary had some stain, be it the *original* or the *current one*. However, since she lacked *actual* sins, it follows that she had *original* sin.[50] In addition, when this thesis says "nothing greater can be understood", "understand" means to think or know *rationally*, and it is not possible to *rationally* think that Mary, born of the voluptuousness of a man, lacks original sin.[51] Thirdly, when this thesis says that this conception was achieved by *a purest mother*, this must be understood in the sense that this supreme purity was realized when Mary conceived the Son of God; then, she was completely purified and remained the purest, from which it cannot be concluded that when she was born she was without original sin.[52]

2.  To the claim to justify Mary's Immaculate Conception based on what St. Augustine says about sins, Bonaventure replies that this saint refers to *current* sins, not *original sin*.[53]

3.  Faced with the allegation of the immaculists that the honor of the Mother refers to the Son, our author responds that although this is true, it does not follow from this that all the honors attributable to the Son must be attributed to the Mother: in his opinion, this would not be honoring the Son but insulting him by attributing to another an honor that only belongs to him. And, as the honor of being exempt from all sin, both current and original, is only proper to the Son of God, since he was conceived of the Holy Spirit and born of a Virgin mother, that honor should not be attributed to Mary. The other dignities are enough for her, with which she surpasses all human praise; in addition, the Virgin does not need false honors, since those that conform to the full truth are enough for her.[54]

4.  Confronting the argument of the defenders of Mary's Immaculate Conception stating that between the extremes it is necessary to place a middle ground, Bonaventure answers that this would be true if this middle was produced by the extremes according to a certain order and congruence. But that is not the case here, since it is not convenient to put a cause without an effect, or an effect without a cause.[55]

5.  To the allegation that the Virgin's sanctification completely surpasses that of the other saints both in magnitude and speed, our author replies that although this is true, it does not follow from this that the Virgin was sanctified in the first moment of the creation of his soul, since she only excels in what is her own.[56]

6.  Lastly, our author discusses the objection of the defenders of Mary's Immaculate Conception, for whom it was possible that grace was infused into the Virgin in the first moment of her creation because, since "nothing was impossible for God", it could make the Virgin immune to all sin. The Seraphic Doctor responds to this allegation by saying: it is appropriate that this privilege should be granted only to him through whom the salvation of all was produced, that is, Jesus Christ, *so that no flesh may glorify before him, but only to him honor and glory would be rendered forever and ever*.[57]

Ultimately, it is necessary to recognize the outstanding honesty with which Saint Bonaventure justifies with numerous arguments his position contrary to the immaculist thesis, which would end up being imposed as an official dogma in 1854. By assuming this position, the Seraphic Doctor was not content to gregariously uphold the most widespread opinion at the time. Rather, he took pains to scrutinize any reasoning for and against the immaculist belief before carefully proposing his own arguments against this belief. If we wanted to present in a brief (very impoverishing) synthesis Bonaventure's arguments to oppose the immaculist doctrine, we could summarize them in these three essentials: (1) the Virgin Mary, having been conceived by her parents through sexual intercourse—which, at that time, was considered essentially linked to original sin—was not immune from this sin; (2) had she been conceived without original sin, Mary would have been equated with Christ, who is the only sinless person; (3) had she had the privilege of an Immaculate Conception, Mary would not have received the gift of Redemption from sin, which Christ comes to grant to all human beings.

### 2.3.2. Arguing the Virgin Mary's Sanctification before Her Birth

In the Third Question of the Third Distinction, Part I, Article I of the *Third Book of Comments on the Sentences of Peter Lombard*, Bonaventure studies the problem of whether the Virgin Mary was sanctified before her birth.[58] Here again, following the scholastic methodology, our author exposes the reasons in favor of this thesis and then the reasons against it before providing his own position on the matter. The Seraphic Doctor proposes three reasons in favor: the first is the emphatic statement of Saint Bernard when he says that "the mother of God was without a doubt holy before birth."[59] The second is a new sentence of Saint Bernard, when he says that "What is known to have been granted to a few saints, it is not fair to believe that it has been denied to the Virgin"; if Jeremiah and John the Baptist were sanctified in the womb, it seems much more certain that the Virgin Mary was sanctified in the womb.[60] As a third argument in favor of this thesis, Bonaventure

highlights the fact that the Church solemnly celebrates the nativity of the Virgin and, since it would not be fair to celebrate this liturgical feast if she had been born in sin, it follows that Mary was sanctified at birth.[61] Then, the Seraphic Doctor adds this disquisition, which is actually a fourth argument in favor of Mary's sanctification before her birth: if any person should have participated in the fullness of the grace of the Virgin's Son, with the greatest right his Mother Mary should have participated in that grace because Mary was most suitable for that grace, and with her and in her that grace increased by being gestated in her mother's womb. It seems logical that Mary was filled with the Holy Spirit's grace in the very womb of her mother Anna.[62]

The Seraphic Doctor then exposes the arguments against the thesis of Mary's sanctification before her birth. The first, glossing the epistle of Saint Paul to the Romans, our author says that if Christ was the first born without sin and the first to rise impassively, it turns out that Mary, having been born before Christ, was not born without sin, and therefore, she was not sanctified before birth.[63] In the second argument against the thesis of Mary's sanctification before her birth, the author refers to the affirmation of Saint Augustine, according to which the grace through which we become temples of God is only given to the reborn, who are or can be born human beings; therefore, as the Virgin was made a temple of God by her sanctification, she was not holy before she was born.[64] Bonaventure raises the third proof contrary to the thesis of Mary's sanctification before her birth in these terms: no property disappears while it has continuity with its cause but, since the soul contracts original sin through parents, it seems clear that while the offspring is united to its mother, it cannot be cleansed of original sin; therefore, it too cannot be sanctified.[65]

The fourth argument against this thesis is based on the principle that *being ordered* presupposes *being different*; since the grace of sanctification presupposes being different, as nothing can be conferred on a being without a discrete or separate existence, therefore, the progeny, while it is united to its mother (without having a being separate from her), cannot be sanctified.[66]

In the fifth argument against this latter thesis, Bonaventure considers that *sanctification in the womb* is an effect known only to God and that no saint is said to have been sanctified in the womb except Jeremiah and John the Baptist, the two mentioned expressly in Scripture; now, since nothing of such nature is said about the Virgin, it seems logical to conclude that Mary was not sanctified before birth.[67]

Having analyzed the arguments for and against the thesis under study, Bonaventure finally exposes his position on the matter, joining the defenders of the sanctification of the Virgin before her birth. In this sense, he highlights that the Church retains as an indubitable truth that the Virgin was sanctified in the womb. This statement is manifested by the fact that the whole Church celebrates the feast of the birth of Mary, something that would be impossible to do if she had not been sanctified before birth.[68]

Bonaventure adds that although we are certain that the Virgin was sanctified in the womb before she was born, we cannot know with certainty which day and at what instant she was sanctified. Our author concludes that Mary's sanctification or infusion of grace occurred shortly after the infusion of the soul in her body.[69]

Finally, in a long and complex disquisition, the Seraphic Doctor assures that if Jeremiah and John the Baptist were sanctified in the wombs of their mothers before being born, it stands to even greater reason the Virgin Mary had to be sanctified before being born For St. Bonaventure this is evident, especially when considering that the holiness of Mary far exceeds the purity and virginity of Jeremiah and John the Baptist, whose holiness is less than that of Mary. Furthermore, in them only virginity is given, while in Mary, virginity is given with motherhood, and if both saints were sanctified in the wombs of their mothers, it stands to greater reason that this Virgin who gestated God in her own womb should be sanctified in the womb of her mother.[70]

### 3. Conclusions

St. Bonaventure addresses with exemplary honesty the arduous and controversial issue of whether the Virgin Mary was conceived without original sin. With such a moral attitude, he systematically examines with great objectivity each of the arguments in contention, both those offered by the defenders of Mary's Immaculate Conception and those offered by the opponents of such a belief. Only after this detailed analysis of the arguments for and against this thesis does the Seraphic Doctor expose the variety of reasoning with which he justifies his own opinion on the matter.

Thus, in accordance with the majority position at that time and based on numerous arguments, among which are several extracted from the Scripture and testimonies of some saints, Bonaventure denies the Immaculate Conception of Mary. According to him, all human beings except Jesus Christ are conceived through carnal intercourse and are therefore born contaminated with the original guilt inherited from our First Parents. In this regard, the Virgin Mary, who was also conceived through carnal intercourse, could not be an exception. Moreover, Christ, the only sinless man, redeemed all humanity from sin, including Mary, who contracted original sin like all other human beings.

However, the Seraphic Doctor defends that although initially contaminated by original sin when conceived through carnal intercourse, the Virgin Mary was sanctified in the maternal womb shortly after the animation of her body (the infusion of the soul into her body) and before her birth. In his opinion, this is evident since if John the Baptist and Jeremiah were sanctified in the womb, Mary, who is far superior to both, should have been sanctified shortly after her body received the animation of her soul.

In the end, many centuries later, the Church would reject the opinion of St. Bonaventure, together with the opinions of the other deniers of the Immaculate Conception of Mary, which would be defined as a dogma in 1854 by Pope Pius IX through the papal bull *Ineffabilis Deus*[71]. To substantiate this dogma doctrinally, the Church took into consideration the arguments put forward in favor of this belief throughout the centuries by countless teachers of Christian doctrine; among them, John Duns Scot, Pietro Aureolo, Francesco de Mayronis, Raymond Llull, the Pope Sixth IV, and Francisco de Santiago stand out for their remarkable roles as fervent defenders of the immaculist belief. Despite this disavowal, one must acknowledge the commendable intellectual integrity with which the Seraphic Doctor attempted to provide a credible answer to this challenging and hotly debated doctrinal problem.

**Funding:** This research received no external funding.

**Institutional Review Board Statement:** Not applicable.

**Informed Consent Statement:** Not applicable.

**Data Availability Statement:** Not applicable.

**Conflicts of Interest:** The author declares no conflict of interest.

### Notes

[1]   Bonaventura de Balneoregio, *III Sent*, d. 3, a. 1, q. 1–3: Q III, 61a–72b.

[2]   "1. Ieremiae primo: *Antequam farmarem te in utero, novi te*; constat, quod beata Virgo excellentior fuit quam Ieremias: ergo prius fuit approbata et purificata eius caro, quam esset formata. Sed ante fuit formata quam animata: ergo ante fuit carnis sanctificatio quam animatio." (Bonaventura de Balneoregio, *III Sent*, d. 3, p. 1, a.1, q. 1: Q III, 61a)

[3]   "2. Item. super illud Lucae primo: *Exultavit* in *gaudio infans* in utero meo; Glossa; « Nondum erat in eo spiritus vitae, et tam erat spiritus gratiae": ergo multo fortius hoc fuit in Domini matre, quae dignior fuit quam Ioannes; ergo sanctificata fuit ante spiritum vitae, et ita ante animationem." (Ibid.).

[4]   "3. Item, non est minus potens virtus sanctificans, quam sit virtus inficiens et foedans; sed caro infici potest et foedari ante animationem: ergo pari ratione ante infusionem, immo fortiori potest sanctificari, cum "opposita nata sint fieri circa idem". Si ergo beatae Virgini hoc concessum est, quantumcumque congruum est concedi purae creaturae; videtur, quod caro eius ante animationem sanctificata fuerit." (Ibid.).

[5]　"4. Item, conceptio gloriosae Virginis fuit de legitimo matrimonio et de parentibus sanctis et de matre sterili, secundum quod narrat quaedam historia: videtur ergo, quod illa conceptio fuerit virtute Spiritus sancti: si ergo illa conceptio, quae est secundum legitimum matrimonium et secundum Spiritus sancti adiutorium, est sancta et immaculata; videtur, quod caro Virginis Mariae in ipsa sua conceptione fuerit sanctificata. Sed ante fuit conceptio quam animatio: ergo fuit ante sanctificata quam animata." (Ibid.).

[6]　"1. Bernardus ad Lugduneuses: «Ante conceptionem sanctificari minime poterat, quia non erat; sed nec in ipso conceptu propter peccatum. quod inerat"; sed constat, quod illud non est (col. 61a) dictum propter *peccatum*, quod esset in parentibus, quia potuissent eam concipere sine peccato: ergo dicit propter *causam* peccati, quae erat in carne: ergo non videtur, quod santificatio fuerit ante infusionem animae." Ibid., 61a–61b.

[7]　"2. Item, sanctificatio est per aliquod munus gratiae Spiritus sancti; sed gratia non habet esse in carne, sed in anima: ergo non videtur, quod ante fuerit carni gratia infusi quam animae." (Ibid. 61b). Como cuarto argumento SB señala que.

[8]　"3. Item, esse gratiae praesupponit *esse* naturae, et perfectio gratiae perfectionem naturae: si ergo sanctificatio dicit effectum gratiae vivificantis, impossibile videtur, quod caro sanctificetur, antequam ab anima vivificetur." (Ibid.).

[9]　"4. Item, nihil pertinet ad sanctificationem, quod non pertinet ad glorificationem; sed caro ante animationem non pertinet ad glorificationem nec resurrectionem [ . . . ]—igitur caro ante animationem non erat idonea ad sanctificationem: ergo non fuit caro Virginis ante sanctificata quam animata." (Ibid.).

[10]　"Respondeo»: Dicendum, quod caro beatae Virginis ante animationem non fuit *sanctificata:* non quia Deus non potuerit carnem Virginis *purificare* ante quam animare, sed quia *sanctificatio* habet esse per aliquod donum gratuitum superadditum, quod quidem non habet esse in carne, sed in anima. Ideo si caro beatae Virginis dicitur *sanctificari*, aut hoc intelligitur medíante gratia existente in *eius anima*, vel mediante gratia existente in *animabus parentium.* Constat, quod non mediante gratia existente in *eius anima*, quia tunc esset oppositio in adiecto, (col 61b) videlicet quod caro sanctificetur ante animae creationem, et tamen sanctificetur per virtutem gratiae illius animae; sequitur enim quod idem sit prius et posterius in uno et eodem." (Ibid., 61b–62a).

[11]　"Si autem intelligatur hoc fieri per gratiam, quae collata fuerit *animabus parentum*, hoc non potest esse, triplici ratione.Prima: quia gratia sanctificationis non habet transfundi a parente in prolem, pro eo quod proles non est in parente secundum animam; ideo nec in parente habet sanctificationis gratiam. [ . . . ] exponendum est secundum animam; erat enim in eis secundum carnem causaliter, sed illud me non sufficiebat ad sanctificationis gratiam, quae respicit animam." (Ibid., 62a).

[12]　"Secunda ratio est: quia, etsi hoc esset possibile, quod sanctificatio derivaretur a parente in prolem, sicut originalis iustitia; nunquam tamen derivatur mediante coitu libidinoso, quia tunc duo opposita essent simul et semel in eodem. Et ideo dicit Bernardus: «Forte inter amplexus maritales sanctitas ipsi conceptioni se immiscuit, ut simul sanctificata fuerit et concepta. Sed hoc ratio non admittit. Quomodo namque sanctitas absque Spiritu sanctificante, aut Spiritui sancto societas cum peccato fuit? Aut quomodo peccatum non fuit, ubi libido non defuit»?" (Ibid.).

[13]　Tertia ratio est: quia esto quod sanctificatio adesset, et libido defuerit virtute divina; non tamen decuit, ut deesset, propter hoc quod haec est solius beatae Virginis praerogativa; sola enim ipsa, ut Sancti dicunt, sine peccato concepit et sine dolore peperit; et ideo hoc parentibus concedi non debuit beatae Virginis, sed soli Virgini reservari. Unde Bernardus: «Dico, Virginem gloriosam de Spiritu sancto concepisse, non autem conceptam fuisse; dico, peperisse virginem, non tamen partam a virgine. [ . . . ] Et ideo simpliciter concedendum, quod caro eius ante animationem non fuit sanctificata. Et concedendae sunt rationes, quae hoc probant." (Ibid., 62a–62b).

[14]　"1. Ad illud ergo quod obiicitur in contrarium de Ieremia: Ante cognovit, quam formavit; dicendum, quod notitia illa non ponit aliquid circa ipsum, sed circa propositum Dei praedestinantis, sicut electio et dilectio et praedestinatio aeterna non ponit aliquid circa praedestinatum; et ideo non sequitur ex hoc, quod caro Ieremiae vel Virginis ante fuerit sanctificata quam formata." (Ibid.).

[15]　"2. Ad illud quod obiicitur de Ioanne, quod nondum erat in ipso spiritus vitae; dicendum, quod in Scriptura aliquid dicitur *fieri*, quando *innotescit*; et *non esse*, quando *non apparet*. Et ideo dicitur spiritus non fuisse in Ioanne, quia non apparebat; adhuc enim in utero erat; spiritus gratiae in eo dicitur *fuisse*, quando in occursum Domini exsultavit in ventre." (Ibid.)

[16]　"3. Ad illud quod obiicitur, quod caro potest infici ante animationem, ergo et sanctificari; dicendum, quod non est simile, quia generatio carnis ex carne est carnalis, non spiritualis; ideo magis habet transfundi caro vitiosa ex carne vitiosa, quam sancta ex parentibus sanctificatis per gratiam." (Ibid.).

[17]　"4. Ad illud quod obiicitur, quod conceptio Virginis fuit ex legitimo matrimonio, ergo etc.; dicendum, quod totum illud est probabile, videlicet quod: concepta fuerit ex sterili et ex coitu coniugali absque omni *culpa actuali*; non tamen sequitur, quod concepta fuerit absque *causa peccati*, quia ita transfunditur originale ex illis qui generant ex coitu legitimo, sicut qui generant ex coitu adulterino". (Ibid.)

[18]　"Et *si obiiciat*, quod non debuit ibi esse *causa* originalis, quia facta est mediante virtute Spiritus sancti fecundante, et propter fecundationem sterilitatis maternae, quae non fuit ab homine, sed a Deo; dicendum, quod illud non valet, quia hoc similiter posset obiici de Ioanne et Isaac, qui de sterilibus sunt concepti." (Ibid., 62b).

[19]　"Propter quod nota, quod Spiritus sanctus aliquando operatur ut *Spiritus*, aliquando ut *sanctus*. Tunc quidem operatur ut *Spiritus*, quando opus facit supra naturam; tunc ut Spiritus *sanctus* operatur, cum non facit solum opus supra naturam, sed etiam sanctificat illud, supra quod operatur.—Dico igitur, quod in conceptione solius Filii Dei, qui est Sanctus Sanctorum, non solum operatus est ut Spiritus, sed ut Spiritus *sanctus*. In conceptione vero, qua Ioannes est conceptus, sive beata Virgo, operatus est ut *Spiritus* tantum; ideo sola Virgo Maria dicitur concepisse de Spiritu sancto." (Ibid., 63a).

20　Bonaventura de Balneoregio, *III Sent*, d. 3, a, 1, q. 2: Q III, 65a–69b.

21　"1. Anselmus de Conceptu virginali: «Decebat, ut illius conceptus fieret de matre purissima ea puritate, qua maior sub Deo nequit intelligi»: sed maior est puritas, ubi nec actualis nec originalis culpa invenitur, quam ubi est aliqua earum: si igitur beata Virgo fuit purissima, non videtur, quod contraxerit originalem culpam: ergo videtur, quod ante originalem culpam sanctificata fuerit." (Ibid., 65a).

22　"2. Item, Augustinus de Natura et gratia: «De Matre Domini, cum de peccatis agitur, nullam volo prorsus habere quaestionem»: ergo non videtur in ea fuisse peccatum, nec actum nec contractum." (Ibid.)

23　"3. Item, ratione videtur: «Honor Matris refertur ad Filium», ut dicit Damascenus; sed omnis honor debetur Filio gloriosae Virginis, maxime quantum ad immunitatem peccati: si igitur ad honorem Filii spectat habere matrem puram et sanctam, videtur, quod Mater Christi immunis fuerit ab omni culpa, tam originali qnam actuali." (Ibid.)

24　"4. Item, si est ponere extrema, contingit ponere medium; sed contingit ponere habentes originale peccatum in anima et in carne, sicut filii Adam habent communiter; contingit invenire carentem in anima et came, sicut in Christo: ergo contingit reperire medium, scilicet habentem in carne et non in anima, vel in anima et non in carne. Sed habere in anima et non in carne est impossibile, quia originale ortum habet a carne; anima enim non traducitur, sed caro: restat igitur, quod aliqua persona fuit, quae haberet originale solum in carne et non in anima; haec autem fuit illa, quae maxime accedit ad Christum puritate, et haec est Virgo Maria: ergo etc." (Ibid., 65b).

25　"5. Item, sanctificatio beatae Virginis excellit sanctificationem aliorum Sanctorum: ergo cum beatus Ioannes sanctificatus fuerit quantum ad nativitatem *ex utero*, quam Ecclesia celebrat; videtur, quod Virgo Maria non tantum sanctificatan sic fuerit, sed etiam in quantum ad nativitatem *in utero*. Sed nativitas in utero est in animae infusione: igitur beata Virgo in instanti infusionis animae sanctitatem habuit divinae gratiae: ergo nunquam habuit infectionem originalis culpae." (Ibid., 65b–66a).

26　"6. Item, possibile fuit, gratiam dari animae beatae Virginis in primo instanti creationis; sed congruum est ponere, quod animae illi id Deus dederit, quod congruebat ei suscipere: ergo videtur, quod in primo instanti originalem dederit ei gratiam: ergo cum gratia non possit simul stare cum culpa, aut gratiam perdidit, aut originalem culpam non contraxit. Sed non est dicere, quod gratiam perdidit: ergo videtur esse dicendum, quod non habuit culpam originalem." (Ibid. 66a).

27　"1. Ad Romanos quinto: *Omnes in Adam peccaverunt*; hoc autem non est, nisi quia fuimus in Adam secundum seminalem rationem: ergo si Virgo fuit secundum seminalem rationem, videtur, quod contraxerit originale, sicut et al.ii." (Ibid.).

28　"2. Item, Augustinus: «Nemo liberatur a massa peccati nisi in fide Redemptoris »: ergo omnes quotquot eripiuntur, per Christum eripiuntur; sed non liberatur quis a peccato, quod non habet: ergo videtur, quod omnes alii a Christo contraxerint originale peccatum." (Ibid.).

29　"3. Item, Bernardus: «Salvator noster, sicut pro omnibus liberandis venit, ita nullum liberum a reatu reperit», ergo, nec beatam Virginem invenit liberam: ergo originale peccatum contraxit." (Ibid.).

30　"4. Item, hoc ipsum videtur *ratione:* quia, si beata Virgo caruit originali peccato, caruit merito mortis: ergo vel iniustitia facta est ei, cum mortua est, vel dispensative pro salute generis humani mortua est. Sed primum facit ad *contumeliam Dei*, quia, si illud verum est, Deus non est iustus retributor; secundum ad *contumeliam Christi*, quia, si illud verum est, Christus non est sufficiens redemptor; ergo utrumque falsum est et impossibile. Restat igitur, quod habuit peccatum originale." (Ibid., 66a–66b).

31　"5. Item, nullus pertinet ad redemptionem Christi, nisi qui habet culpam: si ergo beata Virgo caruit originali, videtur, quod ad redemptionem Christi non pertineat. Sed magna est gloria Christo de Sanctis, quos redemit: ergo si non redemit beatam Virginem, nobilissima gloria privatur. Si ergo hoc est profanum et impium dicere, videtur etc."(Ibid., 66b)

32　"6. Item, si beata Virgo peccatum originale non habuit, et nulli est clausa ianua nisi merito originalis peccati; videtur ergo, quod si mortua fuisset ante Christum, statim evolasset ad caelum: ergo non videtur, quod ianua omnibus aperta fuerit per Christum; et ita falsum dicit Apostolus, cum dicit ad Colossenses primo: *Placuit ei per ipsum reconciliare omnia, quae in caelis, sive quae in terris sunt. *" (Ibid.)

33　"Conclusio. *Cum opinione tunc communi auctor putat esse probabilius, beatae Virginis sanctificationem fuisse post originalis peccati contractionem.*" (Ibid.).

34　"Ad praedictorum intelligentiam est notandum, quod quidam dicere voluerunt, in anima gloriosae Virginis gratiam sanctifi-cationis praevenisse maculam peccati originalis.—Nationem autem huius assignant: quia decebat animam gloriosae Virginis sanctificari excellentissime super animas aliorum Sanctorum, non solum quantum ad *abundantiam* sanctitatis, sed etiam quantum ad *accelerationem* temporis; ideo in instanti suae creationis fuit sibi gratia infusa, et in eodem instanti anima infusa est carni." (Ibid.).

35　"Sed quia *omnium mobilium mobilior est sapientia*, et « nescit tarda molimina Spiritus sancti gratia », et multo potentior est gratia quam natura; hinc est, quod effectus gratiae sanctitatis magis praevaluit in carnem quam effectus foeditatis in animam; et ideo culpam non contraxit." (Ibid. 66b–67a).

36　"Haec autem positio videtur posse fulciri multiplici congruentia, tum propter *Christi praecipuum honorem*, quem decebat de matre purissima fieri; tum propter *Virginis praerogativam singularem*, quae debuit in dignitate sanctificationis ceteros Sanctos et Sanctas praeire; tum etiam propter *ordinis decorem*, ut, sicut fuit persona immunis ab originali et in carne et in anima, sive in causa et in

effectu, et persona utroque modo habens originale, sic esset persona media, quae quodam modo haberet et quodam modo non haberet; et ista est beata Virgo, quae mediatrix est inter nos et Christum, sicut Christus inter nos et Deum." (Ibid., 67a).

37 "Et hoc dicunt sonare verbum Anselmi, cum dicit, quod beata Virgo purissima fuit «ea puritate, qua maior sub Deo nequit intelligi». In hoc enim notat, gradum suae puritatis inferiorem esse respectu Filii, et superiorem respectu aliorum Sanctorum." (Ibid.).

38 "Et ideo quasi mediam rationem huius multiplicis congruentiae voluerunt quidam apponere, addentes insuper illud, quod non repugnat *veritati sacrae Scripturae et fidei Christianae Veritati*, inquam, non repugnat, immo potius consonat, si quis eius mysteria attendat." (Ibid).

39 "Beata enim Virgo Maria signifícata fuit per *arcam*; anima vero beatae Virginis signifícata fuit per *urnam*, in qua positum fuit manna. Cum ergo *urna* illa prius fuerit impleta manna quam posita in *arca*, prius, saltem per naturam, sanctificata fuit anima beatae Virginis quam carni unita." (Ibid.)

40 "*Fidei* etiam *christianae*, ut dicit positio praedicta, non repugnat, pro eo quod dicunt, ipsam Virginem ab originali peccato liberatam per gratiam [ . . . ]. *Et iterum*, licet illa gratia praeveniret *animae infectionem*, non tamen praevenit *carnis foeditatem*. Et ideo ratione illius *foeditatis* iuste remanserunt in Virgine poenalitates; gratia enim sanctificationis non obviat *poenae*, sed *culpae*." (Ibid.).

41 "Et hinc est, quod beata Virgo poenalitatibus fuit obnoxia et per Christum liberata a peccato originali, sed aliter quam alii. Nam alii post casum erecti sunt, Virgo Maria quasi in ipso casu sustentata est, ne rueret, sicut exemplum ponitur de duobus cadentibus in luto.—Et per istam viam effugiunt auctoritates et rationes, quae contra eos adducuntur, dicentes, non concludere, quod beata Virgo habuit infectionem originalis peccati quantum ad *effectum*, sed quantum ad *causam solum.*" (Ibid.).

42 "Aliorum vero positio est, quod sanctificatio Virginis subsecuta est originalis peccati contractionem; et hoc, quia nullus immunis fuit a culpa originalis peccati nisi solum Filius Virginis." (Ibid.).

43 "Sicut enim dicit Apostolus ad Romanos tertio: *Omnes peccaverunt et egent gloria Dei*; ibi Glossa: «*Omnes* peccatores invenit gratia Christi, qui solus sine peccato venit, et *omnes egent, gloria Dei*, id est, ut ipse liberet, qui potest; non tu, qui liberatione indiges»." (Ibid.).

44 "Et hoc ipsum dicit Augustinus super Ioannem tractans illud verbum: *Ecce Agnus Dei*; ubi dicit, quod «solus peccata mundi potuit auferre, qui solus sine peccato venit, quia omni peccato caret»." (Ibid.).

45 "Hic autem modus dicendi *communior* est et *rationabilior* et *securior*. *Communior*, inquam, quia fere omnes illud tenent, quod beata Virgo habuit originale, cum illud appareat ex multiplici ipsius poenalitate, quam non est dicere ipsam passam esse propter aliorum redemptionem; quam etiam non est dicere per assumtionem habuisse, sed per contractionem." (Ibid.).

46 "*Rationabilior* etiam est, quia *esse* (col. 67b) naturae praecedit *esse* gratiae, vel tempore vel natura; et propterea dicit Augustinus, quod «prius est nasci quam renasci»; sicut prius est *esse* quam *bene esse*; prius est igitur animam uniri carni, quam gratiam Dei sibi infundi. Si igitur caro illa infecta fuit, ex sua infectione nata erat animam culpa originali inficere. Necessarium est igitur ponere, quod ante fuerit originalis culpae infectio quam sanctificatio." (Ibid. 67b–68a).

47 "*Securior* etiam est, quia magis consonat *fidei pietati* et *Sanctorum auctoritati*. Magis concordat *Sanctorum auctoritati*, pro eo quod communiter Sancti, cum de materia ista loquuntur, solum Christum excipiunt ab illa generalitate, qua dicitur: *Omnes peccaverunt in Adam*. Nullus autem invenitur dixisse de his quos audivimus auribus nostris, Virginem Mariam a peccato originali fuisse immunem." (Ibid., 68a).

48 "*Pietati* etiam *fidei* magis concordat, pro eo quod, etsi Mater habenda sit in reverentia, et magna erga ipsam habenda sit devotio, multo maior tamen est habenda erga Filium, ex quo est ei omnis honor et gloria. Et ideo, quia hoc spectat ad excellentem dignitatem Christi, quod ipse est omnium *Redemptor* et Salvator, et quod ipse omnibus *aperuit ianuam*, et quod ipse *unus pro omnibus mortuus est*; nullatenus ad hac generalitate beata Virgo Maria excludenda est, ne, dum Matris excellentia ampliatur, Filii gloria minuatur". (Ibid.).

49 "Huic igitur positioni adhaerentes, propter honorem Iesu Christi, qui in nullo praeiudicat honori Matris, dum Filius Matrem incomparabiliter antecellit, teneamus, secundum quod communis opinio tenet, Virginis sanctificationem fuisse post originalis peccati contractionem." (Ibid.)

50 "1. Ad illud vero quod obiicitur in contrarium, quod Virgo purissima fuit; dicendum, quod tria attendenda sunt in illa auctoritate: primo, quia dicit *sub Deo*; et in hoc innuit, quod defuit a puritate Christi, et ideo aliquam maculam habuit, vel (col. 68a) *originalem* vel *actualem*. Non *actualem*, ut patebit: ergo *originalem*. Ideo vult dicere, quod caruit omni macula actuali, non originali." (Ibid., 68a–68b).

51 "Aliud attendendum est, quod dicitur, quod *maior nequit intelligi*, scilicet *rationabiliter*, illud enim dicitur *intelligi*, quod *rationabiliter* cogitatur vel cognoscitur; illud *non rationabiliter* cogitatur, quod nata de voluptate viri careat originali peccato." (Ibid., 68b).

52 "Postremo attendendum est quod dicit, quod *de matre purissima*, facta est illa conceptio; in quo innuit, quod haec summa puritas est ponenda in ea pro tempore, in quo concepit Filium Dei; et tunc omnino purificata et mundissima fuit, sicut patebit infra; et ideo non sequitur, caruisse originali culpa." (Ibid.).

53 "2. Ad illud quod obiicitur, quod nulla habenda est quaestio, cum agitur de peccatis secundum verbum Augustini; dicendum, quod Augustinus intelligit de peccato *actuali*, non de *originali*, sicut patet ex serie litterae." (Ibid.).

54 "Et quia hic honor, scilicet immunem esse ab omni peccato, tam originali quam actuali, solius Filii Dei est, quia solus conceptus de Spiritu sancto et natus de Virgine; ideo Virgini attribuendum non est. Sufficiunt enim Virgini aliae dignitates, quas sibi

Filius communicavit et dedit, in quibus superexcellit omnes humanas laudes et devotiones; et ideo non oportet novos honores confingere ad honorem Virginis, quae non indiget nostro mendacio, quae tantum plena est veritate." (Ibid.).

55     "4. Ad illud quod obiicitur, quod si est ponere extrema, et medium; dicendum, quod verum est, si medium illud natum sit confici ex extremis secundum debitum ordinem et congruentiam. Non sic autem est in proposito; nam ponere causam sine effectu, vel effectum sine causa non convenit neque decet." (Ibid.).

56     "5. Ad illud quod obiicitur, quod sanctificatio beatae Virginis excellit sanctificationem aliorum Sanctorum quantum ad magnitudinem et celeritatem; dicendum, quod verum est; verumtamen non sequitur ex hoc, quod sanctificata fuerit in primo instanti creationis animae, quia non excellit, nisi in quantum decet." (Ibid.).

57     "Ad illud quod obiicitur, quod possibile fuit, in primo instanti gratiam infundi: dicendum, quod (col. 69a) absque dubio *non erat impossibile apud Deum omne verbum* nec illud, videlicet quod posset facere Virginem ab omni peccato immunem; non tamen hoc decuit alicui concedere nisi ei soli, per quem omnium facta est salus, videlicet Domino Iesu Christo, *ut non glorietur in conspectu eius omnis caro; sed ipsi soli sit honor et gloria in saecula saeculorum. Amen.*" (Ibid., 69a–69b).

58     "Dist. III, p. I. Art. I. Quaestio III. *Utrum beata Virgo sanctificata fuerit ante nativitatem.*" (Bonaventura de Balneoregio, *III Sent*, d.3, 1. 1, q. 3: Q III, 70a–72b).

59     "Tertio quaeritur, utrum beata Virgo sanctificata fuerit ante nativitatem. Et quod sic, videtur. 1. Bernardus ad Lugdunenses: «Fuit procul dubio Mater Dei ante sancta quam nata». (Ibid., 70a).

60     "2. Bernardus: «Quod paucis mortalium constat esse collatum, fas certe non est credere tantae Virgini fuisse negatum»; sed Ioannes et Ieremias sanctificati fuerunt in utero, secundum quod habetur Ieremiae primo: *Antequam exires* etc.; et Lucae primo: *Spiritu sancto replebitur adhuc ex utero matris suae*: ergo videtur multo fortius, quod Virgo Maria fuerit in utero sanctificata." (Ibid.).

61     "3. Item, Ecclesia celebrat et solemnizat Virginis nativitatem; sed statui in peccato solemnitas non debetur: ergo videtur, quod in hora nativitatis fuerit sanctificata." (Ibid.).

62     "Item, si aliqua persona de plenitudine gratiae Filii beatae Virginis debuit participare, Mater eius maxime particeps debuit fieri: ergo sicut tunc maxime erat idonea, ut cum ipsa et in ipsa gratia augmentaretur, cum eam gestabat inter viscera; videtur, quod in ipso utero Spiritus sancti gratia fuisset repleta." (Ibid., 70a–70b).

63     "1. Ad Romanos octavo: Ut sit ipse primogenitus; Glossa: « Quia primus sine peccato natus et primus impassibilis resurgens »: ergo videtur, quod beata Virgo non fuerit nata sine peccato, cum nata fuerit ante quam Christus: ergo non fuit sanctificata ante quam nata." (Ibid., 70b).

64     "2. Item, Augustinus ad Dardanum: « Illa gratia, qua efficimur templum Dei singuli, non nisi renatorum est, quod non nisi nati homines esse possunt »: ergo cum per sanctificationem beata Virgo sit facta templum Dei, non fuit ante sancta quam nata." (Ibid.).

65     "3. Item, nulla proprietas perimitur, quamdiu habet continuitatem cum sua causa: sed anima contrahit originale a parentibus: ergo quamdiu proles iuncta est matri, non videtur, quod ab originali peccato possit mundari: ergo nec sanctificari." (Ibid.).

66     "4. Item, *esse ordinatum* praesupponit *esse distinctum*; sed gratia sanctificationis praesupponit *esse* distinctum: ergo nulli potest conferri nisi habenti *esse* discretum: ergo quamdiu proles coniuncta est matri, non videtur posse sanctificari." (Ibid., 70b–71a).

67     "5. Item, *sanctificari* in utero est effectus cognitus soli Deo, ergo de nullo Sancto asserendum est esse sanctificatum in utero, nisi de eo, de quo expresse legitur in Scriptura tradita a Spiritu sancto; sed hoc non legitur de Virgine: ergo etc." (Ibid., 71a).

68     ""Respondeo: Dicendum, quod pro indubitanti habet hoc Ecclesia, videlicet quod beata Virgo fuerit in utero sanctificata. Et illud patet ex hoc, quod eius nativitatem tota Ecclesia celebrat, quod non faceret, nisi sanctificata esset." (Ibid.).

69     "Si autem quaeratur, qua die vel hora sanctificata fuerit, hoc ignoratur; probabiliter tamen creditur, quod cito post infusionem animae fuerit facta infusio gratiae.—Quidquid tamen sit de hora, pro certo habendum est, quod ante nativitatem sanctificata fuerit." (Ibid.).

70     "In novo habetur de sanctificatione Ioannis, in veteri de sanctificatione Ieremiae; quorum uterque legitur in utero fuisse sanctificatus. Et ex hoc quasi *a minori* colligitur, hoc beatae Virgini fuisse concessum; quia, sicut dicit Bernardus, «quod aliis legitur fuisse collatum, non est credendum Virgini fuisse negatum», maxime cum sanctitas Virginis excedat Ieremiae et Ioannis puritatem et virginitatem. In aliis fuit virginitas, in hac cum virginitate fecunditas. Et si illi sanctificati fuerunt in utero, quia ad Sanctum Sanctorum secundum expressionem prophetiae accesserunt magis, quanto magis illa in utero sanctificari debuit, quae Deum in utero portavit?" (Ibid., 71a–71b).

71     The dogmatic nucleus of the bull *Ineffabilis Deus* states: "For the honor of the Holy Trinity, for the joy of the Catholic Church, with the authority of our Lord Jesus Christ, with that of the Holy Apostles Peter and Paul and with ours: we declare, we affirm and we define that it has been revealed by God, and consequently, that it must be firmly and constantly believed by all the faithful, the doctrine that maintains that the most holy Virgin Mary was preserved immune from all stain of original guilt, in the first instant of her conception, by singular grace and privilege of Almighty God, in attention to the merits of Jesus Christ, savior of the human race."

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



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
