# Peer review of "Saint Bonaventure’s Doctrine on the Virgin Mary’s Immaculate Conception"

_religions, doi:10.3390/rel14070930_

Round 1
Reviewer 1 Report
The article presented for review is an interesting example of the history of dogmas. It confirms their long and sometimes tortuous process of formation up to the formation of the final verdict of the Church's teaching office. Bonaventure's argumentation was sound, confirmed by a deep study of the history of theology, and - although flawed - contributed to the final definition of the dogma of the Immaculate Conception of Mary.
For the benefit of the reader less initiated into the history of dogma, I offer one brief addition to the article's conclusion. It is stated in the article that the dogma of the Immaculate Conception of Mary was definitively defined and given for belief by Pope Pius IX in 1854. It would be good to give the wording of this dogma and to list those theologians whose opinions had a decisive influence on the final form of the dogma.
Author Response
See enclosed file

Reviewer 2 Report
In the article, the author presents Bonaventure's position against Mary's immaculate conception in a very clear and structured way. The author gives us a detailed insight into Bonaventura's opinion and argumentation, which actually correspond to the anthropological, hamartiological and soteriological thinking of that time.
Still, with all the contribution of the article, I would like to give the author a few suggestions for deepening the work, but I leave it up to him whether he will consider them.
1. In fact, the author follows the structure and argumentation of Bonaventure, and thus gives us a more precise insight into the original texts of Bonaventure. On the other hand, the author brings very little of his own contribution to the chosen topic.
The work would have gained more value if the author, in my opinion, had given an evaluation of Bonaventure's position against Mary's Immaculate Conception, especially because many centuries later the Church still proclaimed the dogma of Mary's Immaculate Conception. So, it somehow remains unclear why Bonaventura's thesis would be still important, especially for today's Mariology. The article does not actually have the author's conclusions.
2. In the Introduction, the author lists the scolars who have dealt with this topic so far and believes that none of them have presented it in Bonaventura's entire argumentation. However, the article does not clearly state what exactly the author's contribution is in relation to the mentioned authors. Nor does the author refer to them in the article itself.
3. One small theological inaccuracy: on page 1, line 40, the author talks about the perpetual virginity of Mary s the "third mariological dogma". Then in lines 44-45 he continues and mentions "two others important Mariological theses". According to this, one could conclude that the Catholic Church has five Marian dogmas, not four.
Author Response
See enclosed file
